# Screening for Fabry Disease in Kidney Transplant Recipients: Experience of a Multidisciplinary Team

**DOI:** 10.3390/biomedicines8100396

**Published:** 2020-10-07

**Authors:** Massimiliano Veroux, Ines P. Monte, Margherita S. Rodolico, Daniela Corona, Rita Bella, Antonio Basile, Stefano Palmucci, Maria L. Pistorio, Giuseppe Lanza, Concetta De Pasquale, Pierfrancesco Veroux

**Affiliations:** 1Organ Transplantation Unit, University Hospital of Catania, Department of Medical and Surgical sciences and Advanced Technologies, University of Catania, 95123 Catania, Italy; marialuisa.pistorio@unict.it (M.L.P.); depasqua@unict.it (C.D.P.); pveroux@unict.it (P.V.); 2Cardiology Department Echocardiography Laboratory, Department of Cardiothoracic and Vascular, Policlinico “Vittorio Emanuele”, University of Catania, 95123 Catania, Italy; inemonte@unict.it; 3C.N.R. Institute for Biomedical Research and Innovation-IRIB, Section of Catania, Via P. Gaifami 18, 95126 Catania, Italy; margheritastefania.rodolico@cnr.it; 4Department of Biomedical and Biotechnological Sciences, University of Catania, 95123 Catania, Italy; coronadany@libero.it; 5Department of Medical and Surgical Sciences and Advanced Technologies, University of Catania, Via Santa Sofia 78, 95123 Catania, Italy; rbella@unict.it; 6Radiology I Unit, Department of Medical Surgical Sciences and Advanced Technologies “GF Ingrassia”, University Hospital “Policlinico-Vittorio Emanuele”, 95123 Catania, Italy; Basile.antonello73@gmail.com (A.B.); spalmucci@unict.it (S.P.); 7Policlinico-Vittorio Emanuele”, University of Catania, 95123 Catania, Italy; giuseppe.lanza1@unict.it; 8Oasi Research Institute-IRCCS. Via Conte Ruggero, 73, 94018 Troina, Italy

**Keywords:** kidney transplantation, Fabry disease, Fabry nephropathy, screening, multidisciplinary team, Lyso Gb3, GLA mutation, D313Y, F113L, D165H, S126G

## Abstract

Fabry disease (FD) is a rare cause of end-stage renal disease requiring kidney transplantation. Data on the incidence of unrecognized FD in kidney transplant recipients are scarce and probably underestimated. This study evaluated the incidence of FD in a population of kidney recipients, with a particular focus of the multidisciplinary approach for an early clinical assessment and therapeutic approach. Two hundred sixty-five kidney transplant recipients were screened with a genetic analysis for α-galactosidase A (GLA) mutation, with measurement of α-Gal A enzyme activity and Lyso Gb3 levels. Screening was also extended to relatives of affected patients. Seven patients (2.6%) had a GLA mutation. Two patients had a classic form of FD with Fabry nephropathy. Among the relatives, 15 subjects had a GLA mutation, and two had a Fabry nephropathy. The clinical and diagnostic assessment was completed after a median of 3.2 months, and mean time from diagnosis to treatment was 4.6 months. This study reported a high incidence of unrecognized GLA mutations in kidney transplant recipients. Evaluation and management by a multidisciplinary team allowed for an early diagnosis and treatment, and this would result in a delay in the progression of the disease and, finally, in better long-term outcomes.

## 1. Introduction

Fabry disease (FD) is a rare X-linked lysosomal storage disorder, caused by deficiency of the enzyme α-galactosidase A (GLA), resulting in an accumulation of globotriaosylceramide in tissues, mainly heart, brain and kidney, leading to progressive organ dysfunction [1]. The estimated incidence in the general population varies from ~1:40,000 in males [2] and ~1:20,000 in females [3], but recent pieces of evidence provided a higher prevalence, up to 1:1250, in newborn screening studies [4,5,6,7], suggesting a high prevalence of nonclassical mutations and variants of unknown significance, where the progression of the disease and the response to enzyme replacement therapy (ERT) are unknown [4,5]. So far, more than 1000 GLA gene mutations have been identified, and about 60% of them are missense mutations, resulting in single aminoacid substitutions in the alpha-galactosidase protein [8]. Although a typical genotype–phenotype correlation has not been clearly identified, the type of mutation might be correlated with the clinical presentation and the severity of the disease. There are two main clinical subtypes of FD: the classic form, which presents typically in male patients with < 1% of GLA activity, resulting in a more severe form presenting with severe acroparesthesia, angiokeratoma, hyperhidrosis, corneal and lenticular opacities and progressive renal and cardiac dysfunction; and the late onset variant, in which FD patients have a prevalent cardiac or renal involvement [5,9].

Progressive renal impairment is a pervasive clinical manifestation of Fabry disease. Studies on FD screening in a hemodialysis population found a prevalence of FD ranging between 0.02% to 1.2% [10,11,12,13,14]. Usually, the first signs are represented by glomerular hyperfiltration associated with mesangial cell proliferation, and they appear between 10 and 20 years of age in male patients with the classic phenotype [5,15,16]. With the progression of the disease, the microalbuminuria and albuminuria may be recorded in up to 50% of patients over 35 years and in 100% of patients over 50 years [17], and there is a progression of glomerulosclerosis, finally leading to end-stage renal disease (ESRD).

Kidney transplantation represents the best replacement therapy for patients with end-stage renal disease [18] and is a recommended therapeutic option even for patients with Fabry nephropathy [15].

The incidence of FD among kidney transplant recipients is not well known. A recent review of screening studies performed in the period 1995–2017 [14] found a prevalence of 0.2–0.4% of FD among female transplant recipients and an incidence 0.38–1.12% among male patients. 

In this study, we investigated the incidence of unrecognized Fabry disease in a population of kidney transplant recipients performed at a single institution, with the extension of screening to the relatives of the affected patients, with a particular focus of the multidisciplinary approach for an early clinical assessment and therapeutic approach.

## 2. Materials and Methods

### 2.1. Study Population

Kidney transplant recipients, aged >18 years, who received a graft between 2000 and 2017, with a functioning graft followed at the Organ Transplant Unit of the University Hospital of Catania, Italy, were included in this study. All patients underwent single kidney transplantation with standard procedure and were on a triple immunosuppressive regimen, as previously described [19]. Patients were invited to participate in the study during the regular follow-up visits, and all patients gave written informed consent. The University Hospital Ethical Committee confirmed that the study did not require ethical approval since it conforms to normal clinical practice. The study was performed in accordance with Helsinki Declaration guidelines.

### 2.2. Genetic Analysis, α-Galactosidase A Enzyme Activity and Lyso Gb3 Levels

All patients included in the study underwent a genetic analysis for GLA mutation to confirm the diagnosis of Fabry disease. We also measured α-Gal A enzyme activity and Lyso Gb3 levels in all patients. Determination of α-Gal A enzyme activity was performed by using the dried blood spot technique, as previously described by Chamoles et al. [20]. Blood was spotted directly on the filter paper after lancet finger prick or venipuncture syringe draw. The entire circle was uniformly saturated. Cards were air dried for at least 2 h and stored in sealed plastic bags at 4 °C for up to 1 week. The GLA gene was analyzed by Polymerase Chain Reaction (PCR) and sequencing of the entire coding region and the highly conserved exon-intron splice junctions. The concentration of biomarker lyso-Gb3 in dried blood spots was measured using high performance liquid chromatography (HPLC) and tandem mass spectrometry. All GLA gene analyses, determination of the concentration of α-Gal A enzyme activity and lyso Gb3 levels were performed at Centogene © Laboratories (Rostok, Germany).

GLA mutations were classified following the indication of ACMG recommendations [21]:Pathogenic variants: a well-established disease-causing DNA change, with a strong genotype–phenotype correlation;Likely pathogenic variants: probable cause of the patient’s phenotype;Variants of uncertain significance (VUS): genetic variant with unknown or questionable impact on a particular clinical phenotype;Likely benign: a variant not likely be the cause of the disease/phenotype;Benign: a variant that is not considered to be the cause of the disease/ phenotype.

### 2.3. Clinical Assessment

Patients with GLA mutation were followed by the Fabry multidisciplinary team of the University Hospital of Catania, Italy, which was responsible for the clinical and therapeutic assessment of patients. The Fabry multidisciplinary team of the University Hospital of Catania included more than 30 specialists with specific expertise on Fabry disease, which are actively involved in the diagnosis and management of potential FD patients, with the aim to significantly reduce the time to diagnosis and treatment.

All patients with a GLA mutation underwent a complete genetic screening extended to relatives, an evaluation of kidney functionality and the following investigations:Cardiovascular evaluation, including echocardiography, nailfold videocapillaroscopy with post-occlusive reactive hyperemia, color doppler ultrasound of carotid arteries and magnetic resonance imaging (T1 mapping) of heart;Nephrological evaluation, including doppler ultrasound of kidney and evaluation of the estimated glomerular filtration rate by using the Chronic Kidney Disease Epidemiology Collaboration creatinine equation, microalbuminuria and proteinuria [22];Neurologic investigation, including trans-cranial Doppler ultrasound and magnetic resonance imaging of the brain (T2 mapping) [23,24];Ophthalmologic investigation, including confocal microscopy (Confoscan 4, Fortune Technologies, Italy);Dermatologic examination, including dermatoscopy;Gastroenterological and endocrinological examinations;Pneumological examination;Psychological and psychiatric examinations.

Asymptomatic patients with no relevant signs and symptoms of FD or alterations at imaging were followed on a yearly basis.

Patients with signs and/or symptoms of Fabry disease (acral pain crisis, cornea verticillata, hypertrophic cardiomyopathy, stroke, chronic kidney disease with proteinuria, abdominal pain, etc.) were included in an institutional database and were evaluated for enzyme replacement therapy according to the consensus guidelines [25,26,27], with a home infusion program [28].

## 3. Results

A total of 265 kidney transplant recipients were screened for FD (Table 1).

There were 175 males (66%) and 90 females (34%), with a mean age of 53.6 ± 12.1 years and a mean time from transplant of 6.1 ± 2.3 years. Glomerulonephritis was the most common cause of ESRD (30.9%), followed by autosomal dominant polycystic disease (21.9%), but 106 patients (40%) had an unknown cause of ESRD. The mean α-Gal A enzyme activity was 6.40 ± 9.89 μmol/l/h, while mean Lyso Gb3 was 1.19 ± 0.8 ng/mL.

The screening identified seven (2.6%) kidney transplant recipients with a GLA mutation (Table 2).

Two kidney transplant recipients had a classic form phenotype with Fabry nephropathy: a 52-year-old man underwent kidney transplantation for ESRD of unknown origin. At the time of transplant, the patient presented with a severe left ventricular hypertrophy (LVH). The screening was made two years after transplantation and identified a hemizygous pathogenic mutation (c493G > C [p.Asp165His]). The patient presented also recurrent abdominal pain, and diffuse angiokeratomas. Two months later, the patient began the enzyme replacement therapy with algasidase α. The patient, however, experienced a progressive worsening of the graft function and recurrence of abdominal pain and is now considered for switch to algasidase β. His 20-year-old daughter with the same GLA mutation presented with a significant proteinuria (>0.6 g/24 h) and was initially treated with algasidase α and then with migalastat. 

The second patient was a 58-year-old woman who received a kidney transplant for an ESRD of unknown origin. Early post-transplant follow-up identified a left ventricular hypertrophy so that the patient underwent the screening for Fabry Disease, which identified a heterozygous pathogenic GLA mutation (c.658 C > T [p.R220X]). The patient immediately started enzyme replacement therapy with algasidase β and, at 8-year follow-up, she had a stable renal function and a slight reduction of LVH. One brother and one sister presented the same mutation, but their clinical symptoms were mainly recurrent cerebral events and a mild renal insufficiency. They were both initially treated with algasidase β and then, as a consequence of the enzyme shortage, with algasidase α. They are both in good clinical conditions, with no progression of renal disease at 8-year follow-up. 

Five more kidney transplant female recipients had a GLA mutation: four were considered pathogenic, but the diagnostic assessment did not identify organ damage other than kidney disease, while in one patient the variant was considered benign; finally, no further patients required the enzyme replacement therapy. 

Interestingly, a female patient with the heterozygous variant c.937G > T (p.Asp313Tyr) received her living transplant from the father who carried the same mutation. No further clinical manifestations of FD were encountered in both the donor and recipient. The recipient had a progressive worsening of renal function, but renal biopsy performed 7 years after transplantation showed only a mild interstitial fibrosis and tubular atrophy, suggesting a chronic allograft nephropathy, with no evidence of FD progression on electron microscopy. The donor is in good clinical conditions with a normal renal function 8 years after the donation.

The study was extended to the probands’ families. This screening identified 15 subjects with a GLA mutation. All subjects underwent a complete evaluation for the FD phenotype, and two more patients with a significant proteinuria (>600 mg/dL), a normal renal function and recurrent cerebral events (one patient), were treated with algasidase α and migalastat, respectively. 

The clinical and diagnostic assessment was completed after a median of 3.2 months, and the mean time from diagnosis to treatment was 4.6 months, suggesting that an in-house multidisciplinary approach may increase the rate of diagnosis of FD and significantly reduce the delay in the diagnosis and treatment.

## 4. Discussion

Kidney involvement is a common complication of Fabry disease. Microalbuminuria and proteinuria are present in a high proportion of patients with FD [17] and after the fifth decade of life the evolution toward ESRD is frequent among this population [17]. 

Older studies from large international registries reported an incidence of FD among patients on dialysis ranging between 0.016% [19] and 0.018% [29,30]. However, many recent studies demonstrated that the incidence of GLA mutations in patients with ESRD may be significantly higher [11]. Doheny et al. [14] collected data from 63 screening studies for a total of 51,363 patients, and they found that, among 36,820 (23,954 Males and 12,866 Females) patients on haemodialysis screened, the incidence of FD was 0.42% for males and 0.68% for females. More recently, a study from Argentina investigating the prevalence of FD among 9604 male patients on hemodialysis [31] found a decreased or absent α-Gal-A activity in 24 patients, with a confirmed diagnosis of FD in 22 patients (0.23%). 

Kidney transplantation may be a relevant option for the management of Fabry end-stage nephropathy, although few cases of kidney transplantation for Fabry disease have been reported in literature. The collaborative transplant study, which includes most of the transplant centers worldwide, identified a total of 143 kidney transplantations performed for Fabry disease in the period 1990–2016, and this accounts for only the 0.01% of kidney transplants for rare causes of ESRD [32]. This incidence may be largely underestimated since it considered only patients with a pre-transplant diagnosis of Fabry nephropathy. 

However, very few studies investigated the incidence of unrecognized FD in kidney transplant recipients [12,33,34,35]. Doheny et al. [14] reported an incidence of 0.54% unrecognized GLA mutations in male kidney transplant recipients (11 of 2031 patients). Of these, five (0.24%) had pathogenic mutations, while six had benign/likely benign variants. In contrast, of the 1043 female transplant patients screened, none had a pathogenic mutation, while three had benign GLA variants. Erdogmus et al. [12], in a recent screening study among 301 kidney transplant recipients, identified one male patient with a pathogenic mutation (0.3%) and two female patients with a GLA mutation of unknown significance (0.6%). 

This study evaluated the incidence of unrecognized FD in a population of kidney transplant recipients performed at a single institution. Moreover, the study was extended also to family members of the affected patients. The screening identified an unexpectedly high incidence of previously unrecognized GLA mutations in kidney transplant recipients. Seven kidney transplant recipients (2.6%) reported a GLA mutation: two patients had a classic phenotype with Fabry nephropathy and started the enzyme replacement therapy, while in four patients we found a pathogenic mutation with no clear evidences of FD-related clinical manifestations. Notably, this was the first study identifying pathogenic mutations of GLA gene in female kidney transplant recipients.

Interestingly, one female patient received the graft from his father who had the GLA mutation p.D313Y. This mutation was also encountered in the other four patients, who were asymptomatic with normal LysoGb3 values. This variant is considered of uncertain significance, following the American College of Medical Genetics and Genomics (ACMG) recommendations, and is widely reported in literature to have a low clinical significance [36,37], although there are some reports of patients carrying this variant, presenting with neurological symptoms [38,39].

A recent systematic review [37] evaluating the clinical relevance of D313Y variant, found that most patients with D313Y variant had normal residual enzyme activity and normal Lyso-Gb3 levels. Moreover, the prevalence of this variant was comparable to that of the not-Fabry high-risk populations (kidney and /or heart disease), although a slight increased incidence was found in patients with neurological disorders [37]. 

There are eight reports in literature of kidney transplantation from a living donor with GLA mutation [40,41,42,43]. In one living donor, based on the pathological deposition of GL-3, chaperone therapy was initiated to suppress the progression of organ damage [41]. Among the eight recipients, only two required ERT, and, in the two patients where kidney biopsy was performed, no clear signs of progression of kidney disease was observed at histology [42,43], as observed in our recipient.

The incidence of unrecognized FD in our population was considerably higher than previously reported. This was probably related to the homogeneity of the study group and to the presence of many clusters of FD in our region. Moreover, the study was extended also to patients with a known causal diagnosis of end-stage renal disease, and this probably increased the likelihood of finding a GLA mutation.

This study, for the first time, extended the screening also to the probands’ families and we were able to find 15 relatives of transplant patients with GLA mutations. This allowed for a prompt diagnosis and treatment was started before irreversible organ damage: three relatives started the enzyme replacement therapy due to Fabry nephropathy with proteinuria, and they have a stable renal function at last follow up. Two more patients with cerebral complications are on enzyme replacement therapy in good clinical conditions eight years after diagnosis. 

We found eight patients with the p.F113L variant, which has been associated in literature with the late-onset form of FD, with severe cardiac involvement particularly in homozygosis [35,36]. The risk of clinically relevant kidney and cerebral involvement is not clear because, in most cases, the mutation presents with an incomplete α-galactosidase deficiency [44,45].

In our study, two patients (one male and one female) with the p.F113L variant presented with proteinuria, initial kidney impairment, and cerebral disorder, and they immediately started the ERT. In six female patients, with normal Lyso-Gb3 levels, the complete clinical evaluation was negative for FD-related symptoms.

Clinical diagnosis of Fabry disease may be challenging, because most of the symptoms may overlap with those of other diseases. This is particularly true in patients with ESRD undergoing kidney transplantation, where many transplant-related complications (renal, cardiac, neurological and gastrointestinal) are very similar to FD symptoms, meaning that the diagnosis may be not suspected and largely delayed. 

Data from large registries demonstrated that the median delay in the diagnosis of FD was 14 years in adults and 5 years in children, after a consultation of about 10 specialists with various degrees of experience in this field [31,46], and this has not improved substantially in recent years. Moreover, in most cases the diagnosis is made at an advanced stage of the disease, and this could significantly limit the efficacy of the enzyme replacement therapy.

Since 2008, the University Hospital of Catania is a referral center for Fabry Disease, with a multidisciplinary team with specific expertise in Fabry disease, with the aim of increasing the rate of early diagnosis of FD, and to reduce the delay in treatment. Notably, all the specialists involved in the multidisciplinary team work at the same hospital, avoiding the need for multiple time-consuming and costly visits to different hospitals. The multidisciplinary team evaluated all patients with a GLA mutation found at screening in kidney transplant recipients and in probands’ families, and this allowed for a significant reduction in time for the clinical assessment, which was completed after a median of 3.2 months, with a mean time from diagnosis to treatment of 4.6 months. A timely diagnosis allowed for a prompt treatment, before irreversible organ damage developed and this, in principle, could be associated with better long-term outcomes. 

Graft and patient survival in Fabry patients are comparable with that of non-Fabry patients undergoing a kidney transplantation [47,48,49,50], although Fabry kidney transplant recipients have a higher mortality for cardiac events with functioning grafts as compared to non-Fabry controls [50]. Kidney transplantation is not able to replace the enzyme deficiency, since there is no cross-correction of the enzymatic defect between cells, and most transplanted patients may require enzyme replacement therapy for preventing the progression of the disease in other organs. There are limited data on the ERT in kidney transplant recipients. Two large series [51,52] demonstrated that ERT (algasidase α and algasidase β) in kidney transplant recipients is able to reduce the creatinine clearance decline and the progression of proteinuria compared with non-treated patients, and is also able to delay the progression of left ventricular hypertrophy and reduce cerebrovascular events [51,52]. 

Although the study reported some new and important findings, we are conscious of its limitations: the study sample is relatively small, but the population presented is homogeneous and this reduced the potential bias deriving from different races, populations, type of transplants, and immunosuppressive protocols; moreover, the study did not include all patients undergoing a kidney transplantation at our institution. 

In conclusion, this study reported a high incidence of unrecognized GLA mutations in kidney transplant recipients. Although most variants were potentially pathogenic, most patients did not have clinical symptoms attributable to FD. The screening allowed for identifying many patients with GLA mutations among relatives of kidney recipients, who received their treatment at a very early stage. This would encourage the screening in all kidney transplant populations, particularly in those patients with an unknown cause of ESRD. Evaluation and management by a multidisciplinary team allowed from an early diagnosis and treatment, before the development of irreversible organ damage, and this would result in a delay in the progression of the disease and, finally, in better long-term outcomes. 

## Figures and Tables

**Table 1 biomedicines-08-00396-t001:** Clinical characteristics of the study group (265 kidney transplant recipients).

	N (%)
Male/Female	175/90
Deceased donor	225 (84.9)
Living donor	40 (15.1)
Age (years)	53.6 ± 12.1
Cause of ESRD (n, %)	
Glomerulonephritis	82(30.9)
Polycystic Kidney Disease	58 (21.1)
Diabetes	19(7.2)
Unknown	106(40%)
Waiting list (months)	22 ± 9.6
Pretransplant dialysis (months)	30 ± 12.5
Donor age (years)	52.6 ± 13.4
Immunosuppression (n, %)	
Induction	79 (29.8)
TAC+MPA + Ster	210(79.3)
CYA+MPA + Ster	24(9)
Eve+MPA + Ster	31(11.7)
α-Gal A enzyme activity (μmol/l/h)	6.4 ± 9.8
Lyso Gb3 levels (ng/mL)	1.19 ± 0.2
Serum Creatinine (mg/dL)	1.47 ± 0.6

Data are expressed as mean values± standard deviation. Legend: ESRD: end-stage renal disease; TAC: tacrolimus; MPA: mycophenolic acid; Ster: steroids; CyA: cyclosporine; EVE: everolimus.

**Table 2 biomedicines-08-00396-t002:** Clinical and molecular information of the proband and relatives with mutation in α-galactosidase A (GLA). Normal values of α-galactosidase A activity assayed in whole blood are >2.6 μmol/l/h. Normal values of lyso-Gb3 are <1.8 ng/mL. The cause of end-stage renal disease refers only to kidney transplant recipients (proband).

Patient	Kinship	Gender	Age at Diagnosis	Cause of ESRD	Enzimatic Activity	Lyso Gb3	Mutation in cDNA	Protein Change	Classification of GLA Mutation	Clinical Information	Treatment	Serum Creatinine (Mg/dl)	Follow Up
**1**	**Proband**	F	60	Unknown	1.6	1.2	heterozygous mutation c.337T > C(p.Phe113Leu)	p.F113L	Pathogenic	ESRD. No other Fabry-related symptoms	None	End-stage renal disease	3 years. No new events
	Sister	F	50		1.5	2.6	heterozygous mutation c337T > C (p.Phe113Leu)	p.F113L	Pathogenic	Proteinuria (>2 g/die), Cerebral disorders, mild renal impairmaent	algasidase α	1.1	3 years. Stable renal function. One episode of transient ischemic attack
	Sister	F	61		3.25	1.6	heterozygous mutation c337T > C (p.Phe113Leu)	p.F113L	Pathogenic	Asymptomatic		0.8	3 years. No new events
	Sister	F	63		3.65	1.5	heterozygous mutation c337T > C (p.Phe113Leu)	p.F113L	Pathogenic	Asymptomatic		0.9	3 years. No new events
	Son	M	40		0.3	3.1	heterozygous mutation c.337T > C	p.F113L	Pathogenic	Proteinuria (>600 mg/die)	Migalastat	0.8	3 years. Stable renal function
	Daughter	F	33		3.4	1.5	hemizygous mutation c337T > C (p.Phe113Leu)	p.F113L	Pathogenic	Asymptomatic		0.7	3 years. No new events
	Daughter	F	36		3.5	1.7	heterozygous mutation c337T > C (p.Phe113Leu)	p.F113L	Pathogenic	Asymptomatic		0.6	3 years. No new events
	Daughter	F	36		2.8	1.7	hemizygous mutation c.337T > C (p. Phe113Leu)	p.F113L	Pathogenic	Asymptomatic		0.7	3 years. No new events
	Nephew	F	26		2.9	1.1	heterozygous mutation c.337T > C (p.Phe113 Leu)	p.F113L	Pathogenic	Asymptomatic		0.6	3 years. No new events
**2**	**Proband**	M	52	Unknown	<0.1	4.3	hemizygous mutation c493G > C (p.Asp165His)	p.D165H	Pathogenic	End-stage renal disease, hypertrophic cardiomyopathy, abdominal pain, angiokeratomas	algasidase α	1.7	8 years. Worsening of renal function. Heart dysfunction. Angiokeratomas
	Daughter	F	23		0.9	2.2	heterozygous mutation c493G > C (p.Asp165His)	p.D165H	Pathogenic	Proteinuria (>600 mg/die)	algasidase α, then migalastat	0.9	8 years. Stabilization of renal function
**3**	**Proband**	F	36	Unknown	NA	1.5	heterozygous variant c.937G > T (p.Asp313Tyr)	p.D313Y	Uncertain	ESRD. No other Fabry-related symptoms		2.4	8 years. Worsening of renal function
	Father (living donor)	M	58		2.7		hemizygous variant c.937G > T (p.Asp313Tyr)	p.D313Y	Uncertain	Asymptomatic		0.8	8 years. No new events
**4**	**Proband**	F	44	Unknown	NA	1.4	heterozygous mutation c.376A > G (p.Ser126Gly)	p.S126G	Pathogenic	End-stage renal disease. No other Fabry-related symptoms		0.9	3 years. No new events
**5**	**Proband**	F	58	Unknown	8	3.2	heterozygous mutation c.658 C > T (p.R220X)	p.R220X	Pathogenic	End-stage renal disease, hypertrophic cardiomyopathy	algasidase β	1	8 years. Stabilization of renal fucntion and cardiomyopathy
	Brother	M	53		1.4	3.5	hemizygous mutation c.658 C > T (p.R220X)	p.R220X	Pathogenic	Cerebral disorders	algasidase α	0.8	8 years. No new events
	Sister	F	62		6	3.8	heterozygous mutation c.658 C > T (p.R220X)	p.R220X	Pathogenic	Cerebral disorders	algasidase α	0.8	8 years. No new events
**6**	**Proband**	F	59	Hemolytic-uremic syndrome	4.9	1.3	heterozygous mutation c. 937 g > T (p. D313Y)	p.D313Y	Benign	End-stage renal disease. No other Fabry-related symptoms		0.9	5 years. No new events
	Mother	F	86		5.4	1.2	heterozygous mutation c. 937 g > T (p. D313Y)	p.D313Y	Benign	Asymptomatic		0.7	5 years. No new events
	Daughter	F	22		4.21	1.3	heterozygous mutation c. 937 g > T (p. D313Y)	p.D313Y	Benign	Asymptomatic		0.7	5 years. No new events
	Daughter	F	25		5.8	1.4	heterozygous mutation c. 937 g > T (p. D313Y)	p.D313Y	Benign	Asymptomatic		0.7	5 years. No new events
**7**	**Proband**	F	60	Interstitial nephritis	NA	1.1	heterozygous mutation c.376AA > G (p.Ser126Gly)	p.S126G	Pathogenic	End-stage renal disease No other Fabry-related symptoms		0.9	2 years. No new events
	Sister	F	56		NA	1.3	heterozygous mutation c.376AA > G (p.Ser126Gly)	p.S126G	Pathogenic	Asymptomatic		0.6	2 years. No new events

ESRD: end-stage renal disease.

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
