# Peer review of "Screening for Fabry Disease in Kidney Transplant Recipients: Experience of a Multidisciplinary Team"

_biomedicines, 2020, doi:10.3390/biomedicines8100396_

Round 1

Reviewer 1 Report

This is a very nice article concerning Fabry disease in renal transplant. Also the families have been studied as far as possible. It comes from one center only and the number of patients is not very high. Suprisingly high number of Fabry patients. 2/7 of the mutations are  D313Y which have been thought to be a non-disease mutation.

I think the authors should discuss more about the D313Y- mutation. The number in this population is suprisingly high, is it only a slump. Do the authors have idea how much D313Y is found in healthy Italian population. How many D313Y:s have been found in former studies. Do you have any biopsies of those two patients?

Some small notices. Why material and methods are after discussion. In the table proband 7 is missing but there is proband 8.

Author Response

Reply to reviewer Reviewer 1 comments

This is a very nice article concerning Fabry disease in renal transplant. Also the families have been studied as far as possible. It comes from one center only and the number of patients is not very high. Suprisingly high number of Fabry patients. 2/7 of the mutations are  D313Y which have been thought to be a non-disease mutation.

I think the authors should discuss more about the D313Y- mutation. The number in this population is suprisingly high, is it only a slump. Do the authors have idea how much D313Y is found in healthy Italian population. How many D313Y:s have been found in former studies. Do you have any biopsies of those two patients?

Thank you, very much for your kind and valuable comments that will improve significantly the manuscript.

A brief statement describing the clinical relevance of D313y variant was added to the manuscript (p.9 lines 269-273) . Unfortunately, there are no data on the prevalence of the D313Y variant in healthy Italian population: however, some studies in high-risk populations (not Fabry end-stage renal disease patients) found a similar incidence of D313Y variant.

A kidney biopsy was performed in both kidney transplant recipients and we found a chronic allograft nephropathy in the living donor transplant recipient and a hemolytic-uremic syndrome in the other patient. In both patients there were no signs of Fabry nephropathy

Some small notices. Why material and methods are after discussion.

This was a formatting error. The material and methods section was moved before results.

In the table proband 7 is missing but there is proband 8.

This was corrected

Reviewer 2 Report

In their manuscript, Veroux et al. present interesting and important data on the incidence of unrecognized Fabry disease in patients receiving kidney transplantations. They not only report patient data, including enzyme activity, lyso-Gb3 levels and the kind of mutation in the GLA gene, but also extend their Fabry screening on the close relatives of these patients. There is a small number of points that should be addressed, but besides this the mansucript can be published in its current form.

Points:
1- Table 2: males showing the GLA mutations should be termed hemizygous (Proband 1: son, M 40; Proband 3: Father, M 58; Proband 5: brother M 53) whereas females should be termed heterozygous (Proband 2: daugther F 23).
2- Methods should be moved between Intro and Results.
3- Author contributions, funding, conflicts of interest: "" should be removed.
4- p2,L53: lysosomal storage disorder
5- p3, L92: different font for +/- and standard deviation. Legend:
6- p4,L257: all patient"s" - s missing.

Author Response

Reply to Reviewer 2 comments

In their manuscript, Veroux et al. present interesting and important data on the incidence of unrecognized Fabry disease in patients receiving kidney transplantations. They not only report patient data, including enzyme activity, lyso-Gb3 levels and the kind of mutation in the GLA gene, but also extend their Fabry screening on the close relatives of these patients. There is a small number of points that should be addressed, but besides this the mansucript can be published in its current form.

Points:
1- Table 2: males showing the GLA mutations should be termed hemizygous (Proband 1: son, M 40; Proband 3: Father, M 58; Proband 5: brother M 53) whereas females should be termed heterozygous (Proband 2: daugther F 23).

Thank you very much for your kind and valuable comments that will improve significantly the manuscript. The suggested changes have been made (Table 2)

2- Methods should be moved between Intro and Results.

This was a formatting error. The material and methods section was moved before results section.

3- Author contributions, funding, conflicts of interest: "" should be removed.

All the “” have been removed, as suggested

4- p2,L53: lysosomal storage disorder

the suggested change has been made (p2, L54)

5- p3, L92: different font for +/- and standard deviation. Legend:

the suggested changes have been made (table 1, p4 l164-165)

6- p4,L257: all patient"s" - s missing.

the suggested change has been made (p2,L93)

Round 2

Reviewer 1 Report

The article can be accepted as such.